# Synthesis of 5-Metalla-Spiro[4.5]Heterodecenes by [1,4]-Cycloaddition Reaction of Group 13 Diyls with 1,2-Diketones

Hanns M. Weinert [1] , Christoph Wölper [1] and Stephan Schulz [1,2,]*

1 Institute of Inorganic Chemistry, University of Duisburg-Essen, Universitätsstr. 5–7, 45141 Essen, Germany
2 Center for Nanointegration Duisburg-Essen (CENIDE), University of Duisburg-Essen, Carl-Benz-Straße 199, 47057 Duisburg, Germany
* Correspondence: stephan.schulz@uni-due.de

**Abstract:** Monovalent group 13 diyls are versatile reagents in oxidative addition reactions. We report here [1,4]-cycloaddition reactions of β-diketiminate-substituted diyls LM (M = Al, Ga, In, Tl; L = HC[C(Me)NDipp]$_2$, Dipp = 2,6-$^i$Pr$_2$C$_6$H$_3$) with various 1,2-diketones to give 5-metalla-spiro[4.5]heterodecenes **1**, **4–6**, and **8–10**, respectively. In contrast, the reaction of LTl with acenaphthenequinone gave the [2,3]-cycloaddition product **7**, with Tl remaining in the +1 oxidation state. Compound **1** also reacted with a second equivalent of butanedione as well as with benzaldehyde in aldol-type addition reactions to the corresponding α,β-hydroxyketones **2** and **3**, while a reductive activation of a benzene ring was observed in the reaction of benzil with two equivalents of LAl to give the 1,4-aluminacyclohex-2,4-dien **12**. In addition, the reaction of L′BCl$_2$ (L = HC[C(Me)NC$_6$F$_5$]$_2$) with one equivalent of benzil in the presence of KC$_8$ gave the corresponding 5-bora-spiro[4.5]heterodecene **13**, whereas the hydroboration reaction of butanedione with L′BH$_2$ (**14**), which was obtained from the reaction of L′BCl$_2$ with L-selectride, failed to give the saturated 5-bora-spiro[4.5]heterodecane.

**Keywords:** group 13 diyls; low-valent metal; cycloaddition; aldol addition





## 1. Introduction

Neutral monovalent, six-electron group 13 diyls LM (Al [1], Ga [2], In [3], Tl [4]; L = HC[C(Me)NDipp]$_2$; Dipp = 2,6-$^i$Pr$_2$C$_6$H$_3$) are group 13 analogues of singlet NHC-carbenes. In particular, alanediyl LAl, and gallanediyl LGa have received steadily increasing interest in recent years due to their interesting ambiphilic electronic nature [5–8], resulting from the presence of both a filled donor (HOMO) and an empty (p-type) acceptor orbital (LUMO). As a result, these neutral diyls often exhibit transition metal-like reactivity [9,10]. LAl is more reactive than LGa and has been found to readily undergo oxidative addition reactions with a wide variety of E-X σ-bonds, including thermodynamically very stable C-F bonds [11–13], whereas, LGa often reacted in a more selective way [14].

Cycloaddition reactions of monovalent group 13 diyls have been less explored. Reactions of a sterically demanding gallanediyl with (*p*-tolyl)NN(*p*-tolyl) gave the 1,2-diaza-3,4-dimetallacyclobutane (**I**) (Scheme 1) [15], while reactions of LAl with alkynes afforded cycloaluminapropenes (**II**) [16]. In addition, aluminum pinacolates were obtained from reactions of both Lal [16] and a diamidoalumanyl anion [17] with Ph$_2$CO (**III**) and Ph(CO)CH$_3$ (**IV**), respectively. Similarly, the digallane (dpp-Bian)Ga–Ga(dpp-Bian) (dpp-Bian = 1,2-bis[(2,6-diisopropylphenyl)imino]acenaphthene) reacted with benzaldehyde to the respective 1,2-diphenyl-1,2-ethaneoate adduct (**V**) [18], while its reaction with 3,6-di-tert-butyl-ortho-benzoquinone occurred with oxidation of the Ga (II) atoms and two dpp-Bian dianions to give the mononuclear catecholate **VI** [19], but this reaction most likely doesn't occur via cycloaddition. In contrast, to the best of our knowledge, the formation of [1,4]-cycloaddition products with diketones is limited to trapping experiments of an in situ formed monomeric in compound (I) with benzil derivatives (**VII**) [20].

**Scheme 1.** Compounds formed by cycloaddition reactions of low-valent group 13 compounds. **II–VI**: dipp = -2,6-$^i$Pr$_2$C$_6$H$_3$; **I**: R = -4-CH$_3$-C$_6$H$_5$, Ar' = -C$_6$H$_3$-2,6-(C$_6$H$_3$-2,6-$^i$Pr$_2$)$_2$; **II**: R$_1$ = R$_2$ = -SiMe$_3$, R$_1$ = R$_2$ = -Ph, or R$_1$ = -SiMe$_3$ and R$_2$ = -Ph; **VII**: R$_3$ = -C$_6$H$_5$, 4-MeO-C$_6$H$_4$, or 4-Br-C$_6$H$_4$.

Our general interest in the reactivity of low valent group 13 diyls LM in $\sigma$-bond activation reactions [21–26] as well as of unsaturated main group element compounds in single electron transfer and cycloaddition reactions [27–30] let us now focus on reactions of group 13 diyls LM (M = Al, Ga, In, Tl) with 1,2-diketones. Both the group 13 elements and the substituents of the 1,2-diketones were found to influence the product formation. In addition to the expected [1,4]-cycloaddition reaction to 5-metalla-spiro[4.5]heterodecenes (**1**, **4–6**, **8–10**), we also observed a [2,3]-cycloaddition reaction of the β-diketiminate ligand as well as an activation reaction of a benzoyl group. Furthermore, 5-galla-spiro[4.5]heterodecene **1** was found to undergo aldol addition, and to commemorate Liebig's studies on "Radicals of Benzoic Acid" [31], we isolated compound **3** from the addition reaction of **1** with benzaldehyde.

## 2. Results and Discussion

### 2.1. Synthesis

The reaction of LGa with one equivalent of butanedione proceeded with [1,4]-cycloaddition and formation of the expected 5-metalla-spiro[4.5]heterodecene **1** (Scheme 2), while no defined product was isolated from analogous reactions of LAl and LIn. LAl was found to be too reactive, resulting in the formation of a rather complex product mixture, (Figure S50), while LIn is less reactive under these conditions, resulting in an incomplete conversion of LIn (Figure S51). In both cases, no defined compound could be isolated. In addition, the reaction of LTl with butanedione gave a colorless precipitate, which is presumed to be an insoluble thallium enolate, as well as LH according to in situ $^1$H NMR spectroscopic studies (Figure S52). The reaction of **1** with a second equivalent of butanedione gave the aldol addition product **2**, which was also selectively formed in the reaction of LGa with two equivalents of butanedione. The formation of **1** is most likely kinetically favored due to a lower energy barrier for the cycloaddition reaction compared to the aldol addition, and hence the equimolar reaction mainly gave compound **1**. Moreover, **1** reacted with benzaldehyde to give the aldol addition product **3**. The formation of the aldol addition products as observed in the reactions with LGa is also expected to occur in the reactions with LAl and LIn, explaining the unselectivity of these reactions.

**Scheme 2.** The reaction of LGa with butanedione **1** and subsequent aldol addition reactions with butanedione (**2**) and benzaldehyde (**3**).

In order to kinetically stabilize the [1,4]-cycloaddition products and to prevent aldol addition side reactions, we increased the steric demand of the 1,2-diketone. LAl, LGa, and LIn reacted selectively with acenaphthenequinone to the corresponding [1,4]-cycloaddition products **4–6**, while LTl reacted to the [2,3]-cycloaddition product **7** with the unsaturated ligand backbone (Scheme 3). The isolation of compound **7** was hampered by its low solubility and its tendency to decompose in solution with the formation of LH.

**Scheme 3.** Cycloaddition reactions of LAl (**4**), LGa (**5**), LIn (**6**), and LTl (**7**) with acenaphthenequinone. The reaction with Tl was performed in *n*-hexane at −80 °C.

In addition, reactions of LAl and LGa with benzil afforded compounds **8** and **9**, respectively, while in situ [1]H NMR monitoring of the reaction of LIn and benzil revealed the formation of multiple products (Scheme 4). The [1,4]-cycloaddition product of the reaction with LIn **10** was finally selectively crystallized from a mixture of acetonitrile and benzene. In contrast, LTl did not react with benzil. This is in agreement with results from quantum chemical calculations, which showed that the energy level of the metal-centered electron lone pair decreases steadily with increasing atomic number, and finally falls below the ligand-centered orbitals in the case of the thallanediyl LTl [32,33].

**Scheme 4.** The [1,4]-cycloaddition reactions of LAl (**8**), LGa (**9**), and LIn (**10**) with benzil.

Similar to the formation of compound **7**, the reaction of LAl with two equivalents of benzil resulted in an addition to the β-diketiminate ligand to give compound **11** (Scheme 5). In marked contrast, the slow addition of benzil to a concentrated solution of two equivalents of LAl proceeded with the activation of one of the two phenyl groups of the benzil molecule to finally give compound **12**. The reduction of a benzene ring demonstrates the high reducing power of the alanediyl LAl.

**Scheme 5.** The reaction of LAl with two and one-half equivalents of benzil resulting in the formation of **11** and **12**, respectivly.

Surprisingly, the reaction of **8** with one equivalent of LAl does not yield **12**, demonstrating that **8** is not a reaction intermediate in the formation of compound **12**. As the E conformation of the benzil molecule is likely prevalent in solution, we suggest that compound **8′** might form in solution as the kinetically-controlled reaction intermediate of the two-electron reduction reaction of benzil with LAl (Scheme 6). **8′** can then either isomerize to compound **8** or reacts with LAl to compound **12** since it shows the correct conformation for the nucleophilic attack of a second alanediyl molecule.

**Scheme 6.** Possible reaction mechanism for the formation of **12** via intermediate **8′**.

We also attempted to extend these studies to reactions of (in situ formed) boranediyl LB. Unfortunately, we failed to synthesize LBCl$_2$ by reaction of LSiMe$_3$ with BCl$_3$, whereas L′BCl$_2$ (L′ = HC[C(Me)NC$_6$F$_5$]$_2$) was quantitatively formed by this approach [34]. The reaction of L′BCl$_2$ with KC$_8$ in the presence of benzil gave the corresponding 5-bora-spiro[4.5]heterodecene **13** (Scheme 7). On the other hand, the hydroboration reaction of butanedione with L′BH$_2$ (**14**), which was obtained from the reaction of L′BCl$_2$ with L-selectride, did not give the saturated 5-bora-spiro[4.5]heterodecane. No reaction took place at room temperature, while decomposition to yet unknown compounds was observed at 80 °C.

**Scheme 7.** The reaction of L′BCl$_2$ with benzil and KC$_8$ (**13**) as well as L-selectride (**14**).

Our studies demonstrate for the first time the ability of group 13 diyls to activate diketones by cycloaddition reaction to give unsaturated dialkolates of the respective group 13 elements. Furthermore, 5-galla-spiro[4.5]heterodecene **1** was found to be a valuable synthon for the synthesis of α,β-hydroxyketones via aldol addition reaction. The decreased reactivity and reduced power of the group 13 diyls with increasing atomic number (LAl > LGa > LIn > LTl) was also demonstrated: LAl was found to reduce one of the benzene rings of the benzil molecule to give the cyclohexadiene derivate **12,** while LTl did not react with benzil. Moreover, in product **7**, the Tl atom remains in the oxidation state +1.

## 2.2. Spectroscopic Characterization and Single Crystal X-ray Structures

The compounds are soluble in benzene and toluene except for compounds **6** and **7**, which were found to be soluble only in tetrahydrofuran and 2-methyltetrahydrofuran. All indium and thallium compounds were found to decompose slowly in solution with the formation of LH. The $^1$H and $^{13}$C NMR spectra of the 5-metalla-spiro[4.5]heterodecenes **1**, **4**, **5**, **8**–**10**, and **13** show comparable resonance patterns for the β-diketiminate ligand L at slightly shifted frequencies as well as the expected signals for the diketones, indicating that all compounds adopt the same symmetrical structures in solution. For compound **10**, this is true in thf, but in benzene, the resonances are significantly broadened, and no assignment to a defined product was possible (compare Figures S28 and S29).

The molecule symmetry in the aldol-addition products **2** and **3** is reduced, resulting in magnetically inequivalent protons of the β-diketiminate ligand L. In addition, the $^1$H NMR spectrum of compound **2** shows broad resonances at room temperature. Temperature-dependent $^1$H NMR spectra showed a reduced number of magnetically equivalent protons, demonstrating dynamic behavior for both **2** and **3** in solution (Figure S5).

IR spectra of **2** and **3** show bands for the CO stretching vibration of the ketones at 1706 and 1702 cm$^{-1}$, respectively. The activation of the benzene ring in compound **12** by the 1,4-addition of two Al centers is indicated by the shielding of the carbon atoms in the $^{13}$C NMR spectrum (39.6 and 32.7 ppm) and the high field shift of the geminal C*H* proton (1.92 ppm) in the $^1$H NMR spectrum.

The molecular structures of compounds **1**, **2**, and **4**–**13** in the solid state were determined by single crystal X-ray diffraction (sc-XRD). Suitable crystals were obtained by fractional crystallization from the reaction solutions except for compound **6**, which were obtained from a solution in 2-methyltetrahydrofuran layered with *n*-hexane. Compounds **1**, **4**, **5**, **6**, **8**, and **11** crystallize in monoclinic space groups, compounds **2** and **10** in orthorhombic space groups, and compounds **7**, **9**, **12**, and **13** in the triclinic space group $P\bar{1}$, respectively (see Figures 1–4 and S53–S57 and Table S1a–d).

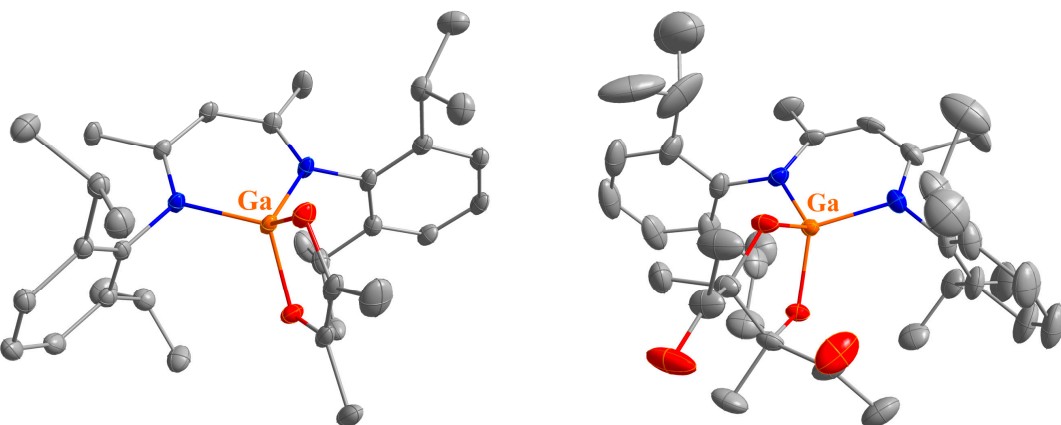

**Figure 1.** Molecular structures of **1** (**left**) and **2** (**right**) in their crystals. Displacement ellipsoids are drawn at the 50% probability level. H atoms are omitted, C atoms are displayed in grey, N atoms in blue, O atoms in red and Ga atoms in orange. Only the major compound of disorders are shown for clarity.

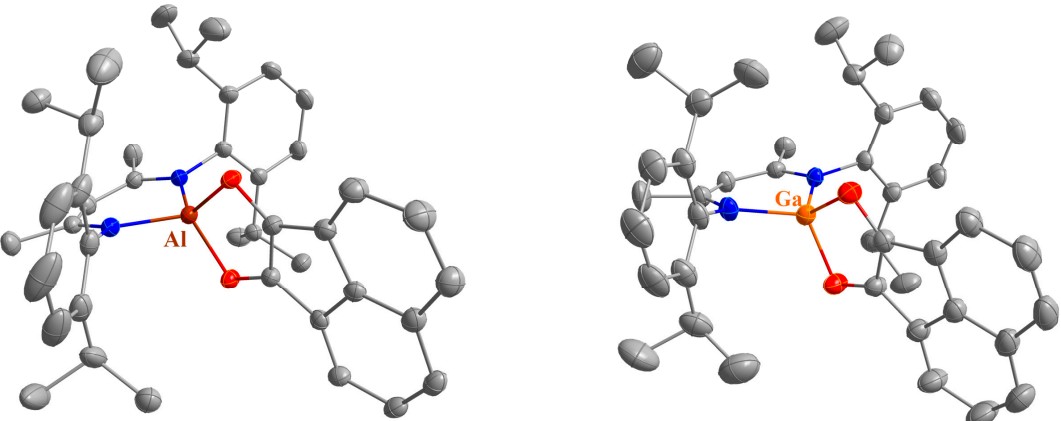

**Figure 2.** Molecular structures of **4** (**left**) and **5** (**right**) in their crystals. Displacement ellipsoids are drawn at the 50% probability level and H atoms are omitted for clarity. C atoms are displayed in grey, N atoms in blue, O atoms in red, Al atoms in dark red, and Ga atoms in orange.

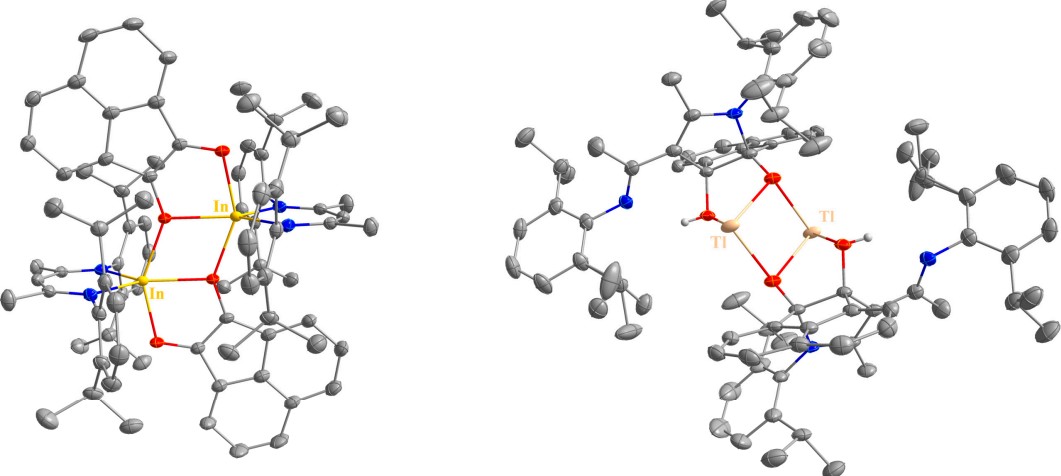

**Figure 3.** Molecular structures of **6** (**left**) and **7** (**right**) in their crystals. Displacement ellipsoids are drawn at the 50% probability level and H atoms (except NH and OH) are omitted for clarity. C atoms are displayed in grey, N atoms in blue, O atoms in red, In atoms in yellow, and Tl atoms in beige.

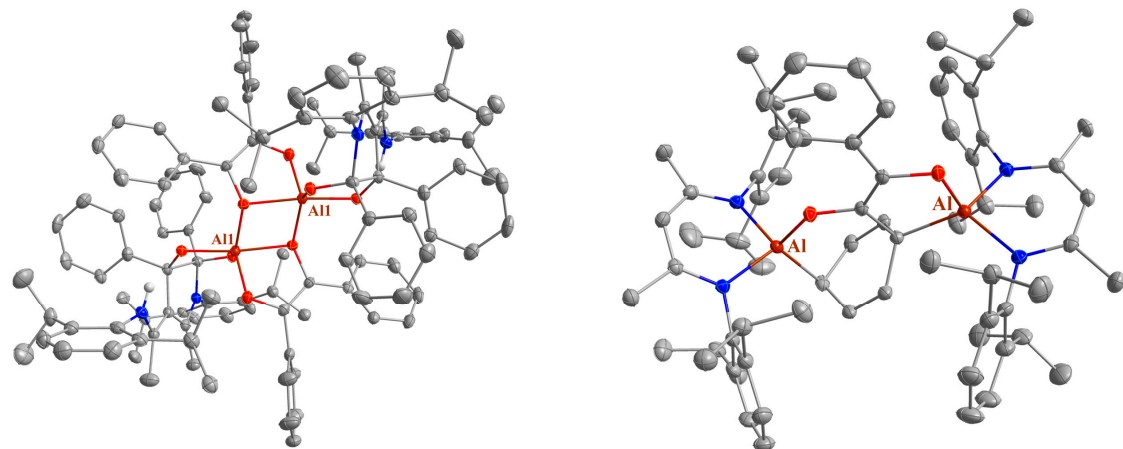

**Figure 4.** Molecular structures of **11** (**left**) and **12a** (**right**) in their crystals. Displacement ellipsoids are drawn at the 50% probability level and H atoms are omitted for clarity. C atoms are displayed in grey, N atoms in blue, O atoms in red, and Al atoms in dark red.

The O–Ga–O bond angles in the five-membered rings of compounds **1** and **2** are 7° more acute than the N–Ga–N angles, see Table 1. In addition, the 5-membered ring in compound **2** is saturated, and only the meso-isomer was found. The 5-metalla-spiro[4.5]heterodecenes (M = B (**13**), Al (**8**), Ga (**9**), In (**10**); see Figures S53–S57) show increasing M–O and M–N bond lengths with an increasing atomic number of the group 13 element, while the O–M–O (105.18(8)° **12**; 93.36(5)°, 92.92(5)° **8**; 91.20(4)°, 91.73(4)° **9**; 82.29(4) **10**) and N–M–N (105.51(8) **13**; 98.21(6), 98.11(7) **8**; 98.99(5)°, 99.79(5)° **9**; 92.23(4) **10**) bond angles decrease. In the case of **10**, the In atom is further coordinated by an acetonitrile molecule, resulting in a penta-coordinated metal center. A dynamic behavior involving the acetonitrile molecule could explain the broad features in the $^1$H NMR of **10** in benzene in contrast to the spectrum in the coordinating solvent thf (the chemical shift of MeCN in thf solution of **10** is found at the frequency expected for solvated MeCN at 1.94 ppm).

**Table 1.** Selected bond lengths [Å] and angles [°] of **1**, **2**, and **4**–**13**.

|  | O–M | O–M–O | N–M | N–M–N |
|---|---|---|---|---|
| **1** | 1.8280(10), 1.8355(10) | 92.89(5) | 1.9122(12), 1.9176(12) | 98.53(5) |
| **2** [a] | 1.827(3), 1.831(3); 1.836(3), 1.831(3) | 91.88(13); 92,45(14) | 1.920(3), 1,916(3); 1.919(3), 1.903(3) | 98.84(15); 98.67(13) |
| **4** | 1.7804(9), 1.7766(8) | 97.18(4) | 1.8582(10), 1.8597(10) | 98.16(4) |
| **5** | 1.8648(14), 1.8665(13) | 94.80(5) | 1.9055(14), 19150(13) | 99.87(6) |
| **6** | 2.0903(9), 2.2310(10) | 79.12(4) | 2.1537(11), 2.1580(11) | 93.39(5) |
| **7** | 2.7463(18) [b], 2.4536(15) [c] | 61.51(5) | / | / |
| **8** [a] | 1.7502(14), 1.7544(14); 1.74958(14), 1.7489(14) | 93.36(5); 92.92(5) | 1.8648(15), 1.8664(14); 1.8641(14), 1.8620(14) | 98.21(6); 98.11(7) |
| **9** [a] | 1.8336(10), 1.8315(10); 1.8293(10), 1.8364(10) | 91.73(4); 91.20(4) | 1.9056(12), 1.9025(11); 1.9050(11), 1.9055(11) | 99.79(5); 98.99(5) |
| **10** | 2.0684(10), 2.0474(10) | 82.29(4) | 2.1192(10), 2.1289(11) | 92.23(4) |
| **11** | 1.7830(16), 1.8087(16); 1.9530(16) [d], 1.7941(17) | 88.15(7); 84.26(7) | / | / |
| **12a** | 1.7393(8), 1.9885(11) [e]; 1.7496(8), 2.0136(11) [e] | 108.90(4)[e]; 94.06(4) [e] | 1.9095(9), 1.9115(9); 1.9125(9), 1.8974(9) | 95.97(4); 95.31(4) |
| **12b** [a] | 1.7332(11), 1.9851(15) [e]; 1.7507(11), 2.014(14) [e]; 1.7355(11), 1.9830(15) [e]; 1.7492(11), 2.0155(15) [e] | 108.17(6) [e], 93.88(5) [e]; 107.81(6) [e], 93.83(5) [e] | 1.9127(13), 1.9022(13); 1.9120(13), 1.9070(13); 1.9148(13), 1.9060(13); 1.9059(13), 1.9151(13) | 95.09(6); 94.83(5); 94.76(5); 95.11(5) |
| **13** | 1.4573(13), 1.4587(13) | 105.18(8) | 1.5622(14), 1.5576(14) | 105.51(8) |

[a]: Two independent molecules in the unit cell; [b]: Tl–OH; [c]: 2.4280(15) Å to the second Tl atom; [d]: 1.8297(15) Å to the second Al atom; [e]: Al–C and O–Al–C instead of Al–O and O–Al–O.

The cycloaddition reaction products obtained from the reactions of acenaphthenequinone with LIn (**6**) and LTl (**7**) are dimeric in the solid state (Figure 3), in contrast to the monomeric 5-metalla-spiro[4.5]heterodecenes **4** and **5** (Figure 2), respectively. This structural difference agrees with the reduced solubility of **6** and **7** compared to **4** and **5** in hydrocarbons and the lower symmetry in solution as indicated by the double number of signals for the $^i$Pr groups (Figure S19). As the NMR spectra of compound **10** in thf-d$_8$ solution retain a high symmetry (Figure S28) it is unlikely that compound **6** dissociates in thf solution to thf-coordinated monomeric species.

The heavier 5-inda-spiro[4.5]heterodecene **6** forms an O-bridged dimer in the solid state with penta-coordinated In centers (Figure 3), while the O-bridged thallium complex **7** has only three-coordinated Tl atoms. The O–M–O bond angles also decrease with the increasing atomic number of M (97.18(4)° **4**; 94.80(5)° **5**; 79.12(4)° **6**; 61.51(5)° **7**) as was observed for compounds **8**, **9**, **10**, and **13**. The two μ-bridging oxygen atoms in compounds **6** and **7** form a rhombus with the two metal centers. Again, the O–In–O angle (69.12(4)°) is more acute than the O–Tl–O angle (83.131(5)°), resulting in almost identical M–M distances (In–In: 3.6082(5), Tl–Tl: 3.6475(3) Å).

The structure of compound **11** represents an intermediate between the structures of compounds **6** and **7**. In **7**, one benzil unit bridges the N and the γ-C atoms as was observed in compound **7**. The second benzil unit forms a 5-membered alumina cycle with a bridging O atom, leading to a dimeric structure in the solid state analogous to that observed for compound **6**, also with a penta-coordinated metal center. However, the Al–Al distance in compound **7** is shorter (2.9442(13) Å) compared to that of compound **6**, and the O–Al–O bond angle (78.36(7)°) falls in between those observed for compounds **6** and **7**.

Finally, two different solvates of compound **12** were structurally characterized with either benzene (**12a**) or *n*-hexane (**12b**) in the unit cell. Surprisingly, a phenyl ring is activated by LAl in compound **12**, resulting in the formation of a 2-alumina-3-oxabicyclo[3.2.2]non-6,8-diene unit. The C–Al–O bond angle (about 108°, see Table 1) in the bicyclic molecule is more obtuse than the angle in the annulated five-membered ring (about 94°), while the Al–C bond length (1.98 Å) is shorter compared to the sum of the covalent bond radii (2.01 Å) [35].

## 3. Conclusions

In this comprehensive study, we demonstrated the general ability of group 13 diyls LM (M = (B), Al, Ga, In) to undergo [1,4]-cycloaddition reactions with diketones to give 5-metalla-spiro[4.5]heterodecenes **1**, **4**–**6**, and **8**–**10** and **13**, respectively. Galla-spiro[4.5]heterodecene **1** also undergoes aldol addition reactions to give α,β-hydroxyketones **2** and **3**. In contrast, thallanediyl LTl failed to undergo cycloaddition reactions at the metal center, but the reaction with acenaphthenequinone proceeded with [2,3]-cycloaddition of the ligand L. Moreover, the higher reactivity of LAl was demonstrated in the reaction of two equivalence LAl with benzil, leading to the activation of a phenyl ring in compound **12**.

## 4. Materials and Methods

All manipulations were performed using standard Schlenk and glovebox techniques under an argon atmosphere, dried by passage through preheated Cu$_2$O pellets and molecular sieve columns. Toluene, *n*-pentane, and *n*-hexane were dried using an MBraun solvent purification system (SPS). Benzene, deuterated benzene, thf, 2-methyltetrahydrofuran and deuterated thf were distilled from Na/K alloy and acetonitrile from CaH$_2$. Solvents were degassed and stored over activated molecular sieves. Starting reagents L'BCl$_2$ [34,36], LAl [1,37], LGa [2,38,39], LIn [3], and LTl [4] were prepared according to the (slightly modified) literature methods (LK was isolated and not prepared in situ). $^1$H (300 MHz, 400 MHz, 600 MHz), $^{11}$B{$^1$H} (128.5 MHz, 192.5 MHz), $^{13}$C{$^1$H} (75.5 MHz, 100.7 MHz, 150.9 MHZ), and $^{19}$F{$^1$H} (376.5 MHz) spectra were recorded with a Bruker Avance DPX-300, a Bruker Avance Neo 400 MHz or a Bruker Avance III HD 600 NMR spectrometer and are referenced to internal C$_6$D$_6$ ($^1$H: δ = 7.16, $^{13}$C: δ = 128.06), and thf-d$_8$ ($^1$H: δ = 3.58;

[13]C: δ = 67.21). Heteronuclear NMR measurements were performed protium decoupled unless otherwise noted. IR spectra were recorded in a glovebox using a BRUKER ALPHAT FT-IR spectrometer equipped with a single reflection ATR sampling module to ensure oxygen- and water-free conditions. Microanalysis was performed at the Elemental Analysis Laboratory of the University of Duisburg-Essen.

### 4.1. Synthesis of LGa($C_4H_6O_2$) (**1**)

A total of 35.3 mg of butanedione (410 µmol) was added to 200 mg of LGa (410 µmol) dissolved in 5 mL of toluene. The clear yellow solution turned orange and was stirred for 15 min. The solution was layered with 15 mL of *n*-hexane and stored overnight at room temperature. Small amounts of solids were separated by filtration and the filtrate was concentrated to about 0.5 mL. The product was obtained as an orange crystalline solid when stored at −30 °C. Yield: 112 mg (197 µmol, 48%).

Anal. Calcd. for $C_{33}H_{47}GaN_2O_2$: C, 69.12, H, 8.26; N, 4.88; Found: C, 69.3, H, 8.29; N, 4.86%. ATR-IR: υ 2960, 2866, 1660, 1587, 1555, 1529, 1462, 1440, 1382, 1316, 1256, 1214, 1176, 1125, 1101, 1022, 984, 932, 872, 803, 772, 761, 747, 668, 533, 533 cm$^{-1}$. [1]H NMR (400 MHz, $C_6D_6$, 25 °C): δ 7.04 (s, 6 H, $C_6H_3$-2,6$^i$Pr$_2$), 4.86 (s, 1 H, γ-C$H$), 3.32 (sept, $^3J_{HH}$ = 6.8 Hz, 4 H, C$H$(CH$_3$)$_2$), 1.74 (s, 6 H, Ga(OCC$H_3$)$_2$), 1.54 (d, $^3J_{HH}$ = 6.8 Hz, 12 H, CH(C$H_3$)$_2$), 1.54 (s, 6 H, ArNCC$H_3$), 1.12 (d, $^3J_{HH}$ = 6.9 Hz, 12 H CH(C$H_3$)$_2$). [13]C NMR (100.6 MHz, $C_6D_6$, 25 °C): δ 171.4 (ArNC CH$_3$), 144.2, 138.8, 128.1, 124.4 (Ar$C$), 131.6 (Ga(O$C$CH$_3$)$_2$), 96.0 (γ-$C$H), 28.7 (CH(CH$_3$)$_2$), 24.8, 24.6 (CH($C$H$_3$)$_2$), 23.3 (ArNC$C$H$_3$), 16.1 (Ga(OC$C$H$_3$)$_2$).

### 4.2. Synthesis of LGa($C_8H_{12}O_4$) (**2**)

8.8 mg of butanedione (103 µmol) was added to 20 mg of LGa (41 µmol) dissolved in 0.5 mL of $C_6D_6$. A colorless precipitate formed shortly after, which was redissolved by heating the suspension. Storage at room temperature gave the product a colorless crystalline solid. Yield: 15 mg (23 µmol, 55%).

Anal. Calcd. for $C_{37}H_{53}Ga_1N_2O_4$: C, 67.38, H, 8.10; N, 4.25; Found: C, 67.0, H, 8.88 N, 4.07%. ATR-IR: υ 2958, 2928, 2866, 1706, 1535, 1464, 1438, 1384, 1347, 1318, 1262, 1217, 1182, 1109, 1082, 1059, 1024, 995, 925, 875, 798, 761, 695, 668, 605, 542, 529 cm$^{-1}$. [1]H NMR (400 MHz, $C_6D_6$, 25 °C): δ 7.15-6.98 (m, 6 H, $C_6H_3$-2,6$^i$Pr$_2$), 4.86 (s, 1 H, γ-C$H$), 3.32 (sept, $^3J_{HH}$ = 6.8 Hz, 1 H, C$H$(CH$_3$)$_2$), 3.18 (m, 3 H, C$H$(CH$_3$)$_2$), 1.96 (s, 4.5 H, GaC$_8H_{12}O_4$), 1.80 (s, 1.5 H, GaC$_8H_{12}O_4$), 1.74 (s, 1.5 H, GaC$_8H_{12}O_4$), 1.53 (m, 18 H, CH(C$H_3$)$_2$, ArNCC$H_3$), 1.12 (d, $^3J_{HH}$ = 6.9 Hz, 3 H, CH(C$H_3$)$_2$), 1.04 (d, $^3J_{HH}$ = 6.9 Hz, 9 H, CH(C$H_3$)$_2$), 0.95 (s, 4.5 H, GaC$_8H_{12}O_{14}$). [1]H NMR (300 MHz, $C_6D_6$, 80 °C): δ 7.06 (s, 6 H, N-$C_6H_5$), 4.94 (s, 1 H, γ-C$H$), 3.31 (sept, $^3J_{HH}$ = 6.9 Hz, 4 H, C$H$(CH$_3$)$_2$), 1.86 (s(br), 6 H, GaC$_8H_{12}O_{14}$), 1.67 (s(br), 1.5 H, GaC$_8H_{12}O_{14}$), 1.63 (s,6 H, ArNCC$H_3$), 1.49 (d, $^3J_{HH}$ = 6.7 Hz, 12 H, CH(C$H_3$)$_2$), 1.14 (d, $^3J_{HH}$ = 6.9 Hz, 12 H, CH(C$H_3$)$_2$). [13]C NMR (100.6 MHz, $C_6D_6$, 25 °C): many resonances were not observed due to line broadening (cp. [1]H NMR spectrum) and low solubility in benzene.

### 4.3. Synthesis of LGa($C_{11}H_{12}O_3$) (**3**)

21.4 µL of benzaldehyde (22.2 mg, 209.3 µmol) was added to 100 mg of LGa($C_4H_6O_2$) (**1**) (174 µmol) dissolved in 1 mL of toluene. A rapid color change from orange to yellow was observed and a precipitate began to form. The suspension was stored in the freezer overnight, filtered, and washed with *n*-hexane, yielding 75 mg of LGa($C_{11}H_{12}O_3$). Yield: 70 mg (111 µmol, 64%).

Anal. Calcd. for $C_{40}H_{53}GaN_2O_3$: C, 70.69, H, 7.86; N, 4.12; Found: C, 70.6, H, 7.96; N, 3.95%. ATR-IR: υ 2960, 2927, 2868, 2814, 1702, 1537, 1438, 1394, 1367, 1359, 1320, 1268, 1182, 1142, 1096, 1051, 1028, 929, 877, 796, 757, 732, 699, 672, 604, 569, 492, 474 cm$^{-1}$. [1]H NMR (400 MHz, $C_6D_6$, 25 °C): δ 7.44, 7.23–7.00 (m, 11 H, $C_6H_3$-2,6$^i$Pr$_2$ and $C_6H_5$), 4.88 (s, 1 H, γ-C$H$), 4.60 (s, 1 H, GaOC$H$), 3.61 (sept, $^3J_{HH}$ = 6.7 Hz, 1 H, C$H$(CH$_3$)$_2$), 3.47 (sept, $^3J_{HH}$ = 6.8 Hz, 1 H, C$H$(CH$_3$)$_2$), 3.15 (sept, $^3J_{HH}$ = 6.8 Hz, 1 H, C$H$(CH$_3$)$_2$), 3.04 (sept, $^3J_{HH}$ = 6.8 Hz, 1 H, C$H$(CH$_3$)$_2$), 1.74 (s, 3 H, GaOC(CO)C$H_3$), 1.61 (s, 3 H, ArNCC$H_3$),

1.60 (s, 3 H, ArNCC$H_3$), 1.59 (d, $^3J_{HH}$ = 6.8 Hz, 3 H, CH(C$H_3$)$_2$), 1.51 (d, $^3J_{HH}$ = 6.8 Hz, 3 H, CH(C$H_3$)$_2$), 1.45 (d, $^3J_{HH}$ = 6.7 Hz, 3 H, CH(C$H_3$)$_2$), 1.26 (d, $^3J_{HH}$ = 6.8 Hz, 3 H, CH(C$H_3$)$_2$), 1.13 (d, $^3J_{HH}$ = 6.9 Hz, 3 H, CH(C$H_3$)$_2$), 1.10 (d, $^3J_{HH}$ = 6.9 Hz, 3 H, CH(C$H_3$)$_2$), 1.05 (d, $^3J_{HH}$ = 6.8 Hz, 3 H, CH(C$H_3$)$_2$), 0.92 (d, $^3J_{HH}$ = 6.8 Hz, 3 H, CH(C$H_3$)$_2$), 0.44 (s, 3 H, GaOCC$H_3$). $^{13}$C NMR (100.6 MHz, C$_6$D$_6$, 25 °C): δ 171.4 (GaOC(CO)CH$_3$), 172.0 (ArNCCH$_3$), 172.0, 144.4, 144.4, 143.9, 143.9, 143.5, 139.7, 139.0, 127.6, 127.1, 126.6, 124.7, 124.7, 124.7, 124.6 (Ar$C$), 95.8 (γ-CH), 84.7 (GaOCCH$_3$), 80.7 (GaOCH), 29.0, 28.9, 28.6, 28.5 (CH(CH$_3$)$_2$), 25.4, 25.0, 24.7, 24.7, 24.5, 24.2, 24.1, 23.7 (CH(CH$_3$)$_2$), 24.6 (GaOC(CO)CH$_3$), 23.4, 23.4 (ArNCCH$_3$), 17.9 (GaOCCH$_3$).

### 4.4. Synthesis of LAl(C$_{12}$H$_6$O$_2$) (**4**)

50 mg of LAl (112 μmol) and 20.5 mg of acenaphthenequinone (112 μmol) were dissolved in 5 mL of toluene and the resulting dark purple solution was stirred overnight at ambient temperature. All volatiles were removed, and the dark purple residue was washed with small amounts of *n*-hexane. Yield: 23 mg (37 μmol, 33%).

Anal. Calcd. for C$_{41}$H$_{47}$AlN$_2$O$_2$: C, 78.56, H, 7.56; N, 4.47; Found: C, 78.1, H, 7.37; N, 4.28%. ATR-IR: υ 2964, 2927, 2868, 1531, 1462, 1444, 1382, 1313, 1249, 1196, 1133, 1105, 1024, 917, 898, 803, 772, 759, 738, 703, 596, 482, 472, 445, 416 cm$^{-1}$. $^1$H NMR (400 MHz, C$_6$D$_6$, 25 °C): δ 7.18 (d, $^3J_{HH}$ = 6.8 Hz, 2 H, AlC$_{12}H_6$O$_2$), 7.10 (d, $^3J_{HH}$ = 8.3 Hz, 2 H, AlC$_{12}H_6$O$_2$), 6.98 (dd, $^3J_{HH}$ = 6.7, 8.3 Hz, 2 H, AlC$_{12}H_6$O$_2$), 6.95-6.87 (m, 6 H, C$_6H_3$-2,6$^i$Pr$_2$), 5.02 (s, 1 H, γ-C$H$), 3.35 (sept, $^3J_{HH}$ = 6.8 Hz, 4 H, C$H$(CH$_3$)$_2$), 1.54 (s, 6 H, ArNCC$H_3$), 1.52 (d, $^3J_{HH}$ = 6.8 Hz, 12 H, CH(C$H_3$)$_2$), 1.10 (d, $^3J_{HH}$ = 6.9 Hz, 12 H CH(C$H_3$)$_2$). $^{13}$C NMR (100.6 MHz, C$_6$D$_6$, 25 °C): δ 172.5 (ArNCCH$_3$), 144.4, 137.2, 128.4, 124.7 (Ar$C$), 146.0, 133.5, 127.4, 127.1, 124.9, 123.4, 116.8 (AlC$_{12}H_6$O$_2$), 98.1 (γ-CH), 28.9 (CH(CH$_3$)$_2$), 24.9, 24.8 (CH(CH$_3$)$_2$), 23.0 (ArNCCH$_3$).

### 4.5. Synthesis of LGa(C$_{12}$H$_6$O$_2$) (**5**)

A total of 50 mg of LGa (103 μmol) and 18.7 mg of acenaphthenequinone (103 μmol) were dissolved in 5 mL of toluene and the resulting dark purple solution was stirred overnight at room temperature. Crystals suitable for sc-XRD were obtained from a highly concentrated solution after storage at −30 °C. However, the product can be more conveniently isolated as a purple powder by removing all volatiles and washing the residue with *n*-hexane. Yield: 35 mg (52 μmol, 51%).

Anal. Calcd. for C$_{41}$H$_{47}$GaN$_2$O$_2$: C, 73.55, H, 7.08; N, 4.18; Found: C, 73.3, H, 7.09; N, 4.49%. ATR-IR: υ 3061, 3034, 2964, 2923, 2866, 1526, 1462, 1442, 1382, 1313, 1256, 1192, 1180, 1131, 1106, 1055, 1024, 921, 900, 883, 800, 769, 759, 627, 596, 581, 526, 472, 441, 420 cm$^{-1}$. $^1$H NMR (400 MHz, C$_6$D$_6$, 25 °C): δ 7.22 (d, $^3J_{HH}$ = 6.6 Hz, 2 H, GaC$_{12}H_6$O$_2$), 7.12 (d, $^3J_{HH}$ = 9.4 Hz, 2 H, GaC$_{12}H_6$O$_2$), 7.22 (dd, $^3J_{HH}$ = 6.6, 8.3 Hz, 2 H, GaC$_{12}H_6$O$_2$), 6.91 (s, 6 H, C$_6H_3$-2,6$^i$Pr$_2$), 4.89 (s, 1 H, γ-C$H$), 3.35 (sept, $^3J_{HH}$ = 6.7 Hz, 4 H, C$H$(CH$_3$)$_2$), 1.54 (s, 6 H, ArNCC$H_3$), 1.51 (d, $^3J_{HH}$ = 6.9 Hz, 12 H, CH(C$H_3$)$_2$), 1.11 (d, $^3J_{HH}$ = 6.9 Hz, 12 H CH(C$H_3$)$_2$). $^{13}$C NMR (100.6 MHz, C$_6$D$_6$, 25 °C): δ 172.1 (ArNCCH$_3$), 144.0, 137.8, 128.5, 124.7 (Ar$C$), 146.6, 134.1, 127.4, 127.1, 124.7, 123.2, 116.7 (GaC$_{12}H_6$O$_2$), 96.5 (γ-CH), 28.9 (CH(CH$_3$)$_2$), 24.8, 24.7 (CH(CH$_3$)$_2$), 23.2 (ArNCCH$_3$).

### 4.6. Synthesis of LIn(C$_{12}$H$_6$O$_2$) (**6**)

200 mg of LIn (376 μmol) and 68.4 mg of acenaphthenequinone (376 μmol) were dissolved in 5 mL of toluene and the resulting dark purple suspension was stirred overnight at ambient temperature. The product was obtained as a purple powder by filtration. Yield: 201 mg (281 μmol, 75%).

Anal. Calcd. for C$_{41}$H$_{47}$InN$_2$O$_2$: C, 68.91, H, 6.63; N, 3.92; Found: C, 68.7, H, 6.54; N, 3.66%. ATR-IR: υ 2967, 2951, 2925, 2866, 1547, 1522, 1458, 1438, 1409, 1382, 1367, 1328, 1316, 1266, 1172, 1130, 1101, 1079, 1055, 1030, 935, 902, 854, 798, 761, 730, 592, 536, 481, 428 cm$^{-1}$. $^1$H NMR (400 MHz, thf-d$_8$, 25 °C): δ 7.52, 7.36, 7.24, 7.07 (m, 6 H, InC$_{12}H_6$O$_2$), 7.18, 7.07, 6.63 (m, 6 H, C$_6H_3$-2,6$^i$Pr$_2$), 5.40 (s, 1 H, γ-C$H$), 3.72 (sept, $^3J_{HH}$ = 6.7 Hz, 2 H, C$H$(CH$_3$)$_2$), 3.09

(sept, $^3J_{HH}$ = 6.8 Hz, 2 H, C$H$(CH$_3$)$_2$), 1.76 (s, 6 H, ArNCC$H_3$), 1.27 (d, $^3J_{HH}$ = 6.6 Hz, 6 H, CH(C$H_3$)$_2$), 1.19 (d, $^3J_{HH}$ = 6.7 Hz, 6 H CH(C$H_3$)$_2$), 0.20 (d, $^3J_{HH}$ = 6.7 Hz, 6 H, CH(C$H_3$)$_2$), −0.28 (d, $^3J_{HH}$ = 6.8 Hz, 6 H CH(C$H_3$)$_2$). $^{13}$C NMR (100.6 MHz, thf-d$_8$, 25 °C): δ 172.7 (ArNC$C$H$_3$), 150.6, 145.2, 144.0, 143.5, 137.2, 135.7, 135.5, 127.6, 127.4, 127.1, 126.9, 126.1, 125.3, 125.2, 123.0, 121.3, 120.0, 116.9 (Ar$C$ + In$C_{12}H_6O_2$), 95.6 (γ-$C$H), 28.5, 28.0 (CH($C$H$_3$)$_2$), 26.4, 25.5, 24.6, 22.2 (CH($C$H$_3$)$_2$), 25.7 (ArNC$C$H$_3$).

### 4.7. Synthesis of LTl(C$_{14}$H$_{10}$O$_2$) (7)

A total of 100 mg of LTl (161 μmol) and 27.8 mg of acenaphthenequinone (153 μmol) were cooled to −80 °C and 2 mL of n-hexane was added. The resulting mixture was warmed to ambient temperature within 8 h, and the resulting off-white solid (60 mg) was separated by filtration. The $^1$H NMR spectrum of this solid is essentially consistent with that obtained from isolated crystals from the reaction mixture, which were characterized by sc-XRD and are highly reproducible between different batches. All attempts to further purify compound **9** failed due to decomposition in the solution state, resulting in the formation of the protonated ligand LH.

$^1$H NMR (400 MHz, C$_6$D$_6$, 25 °C): δ 7.62–6.84 (m, 22 H, C$_6$H$_3$-2,6$^i$Pr$_2$ and TlC$_{12}$H$_6$O$_2$), 4.04, 3.20, 2.57, 1.98 (broad, 4 H, C$H$(CH$_3$)$_2$), 1.91, 1.85 (s, 6 H, ArNCC$H_3$), 1.24, 1.21, 1.20, 1.12, 1.11, 0.74, 0.49, −0.18 (d, 24 H, CH(C$H_3$)$_2$).

### 4.8. Synthesis of LAl(C$_{14}$H$_{10}$O$_2$) (8) and LAl(C$_{14}$H$_{10}$O$_2$)$_2$ (11)

LAl(C$_{14}$H$_{10}$O$_2$)$_2$ (**11**): 100 mg of LAl (225 μmol) and 94.6 mg of benzil (450 μmol) were cooled to −80 °C and 5 mL of *n*-hexane was slowly added and the mixture was warmed to ambient temperature overnight with stirring. The resulting solid was filtered off and dissolved in toluene. Storage at −30 °C afforded 30 mg of LAl(C$_{14}$H$_{10}$O$_2$)$_2$ as a crystalline yellow solid (suitable for sc-XRD). Yield: 40 mg (35 μmol, 16%).

Anal. Calcd. for C$_{57}$H$_{61}$AlN$_2$O$_4$: C, 79.14; H, 7.11; N, 3.24; Found: C, 78.7, H, 7.17; N, 3.23%. ATR-IR: υ 2964, 2928, 2868, 1578, 1529, 1462, 1442, 1371, 1334, 1316, 1253, 1214, 1131, 1057, 1023, 933, 839, 803, 776, 759, 747, 732, 709, 695, 664, 613, 577, 528, 480 cm$^{-1}$. $^1$H NMR (600 MHz, C$_6$D$_6$, 25 °C): δ 11.76 (s, 1 H, ArN$H$CCH$_3$), δ 7.88 (d, $^3J_{HH}$ = 7.7 Hz, 2 H, C$_6$H$_5$), 7.82 (d, $^3J_{HH}$ = 7.7 Hz, 2 H, C$_6$H$_5$), 7.32 (d, $^3J_{HH}$ = 7.7 Hz, 1 H, C$_6$H$_5$), 7.27 (m, 3 H, C$_6$H$_5$), 7.22 (t, $^3J_{HH}$ = 7.6 Hz, 2 H, C$_6$H$_5$), 7.13-6.93 (m, 11 H, C$_6$H$_3$-2,6$^i$Pr$_2$ and C$_6$H$_5$), 6.88 (t, $^3J_{HH}$ = 7.3 Hz, 1 H, C$_6$H$_5$), 6.83 (t, $^3J_{HH}$ = 7.6 Hz, 1 H, C$_6$H$_5$), 6.80 (d, $^3J_{HH}$ = 7.7 Hz, 1 H, C$_6$H$_5$), 6.72 (t, $^3J_{HH}$ = 7.6 Hz, 2 H, C$_6$H$_5$), 6.65 (t, $^3J_{HH}$ = 7.4 Hz, 1 H, C$_6$H$_5$), 6.62 (t, $^3J_{HH}$ = 7.7 Hz, 1 H, C$_6$H$_5$), 6.07 (d, $^3J_{HH}$ = 7.9 Hz, 1 H, C$_6$H$_5$), 4.69 (sept, $^3J_{HH}$ = 6.7 Hz, 1 H, C$H$(CH$_3$)$_2$), 4.45 (sept, $^3J_{HH}$ = 6.7 Hz, 1 H, C$H$(CH$_3$)$_2$), 3.35 (sept, $^3J_{HH}$ = 6.5 Hz, 1 H, C$H$(CH$_3$)$_2$), 2.20 (sept, $^3J_{HH}$ = 6.8 Hz, 1 H, C$H$(CH$_3$)$_2$), 2.11 (s, 3 H, C$_6$H$_5$C$H_3$), 1.92 (d, $^3J_{HH}$ = 6.6 Hz, 3 H, CH(C$H_3$)$_2$), 1.78 (s, 36 H, ArNCC$H_3$), 1.69 (s, 3 H, ArNCC$H_3$), 1.47 (d, $^3J_{HH}$ = 6.8 Hz, 3 H, CH(C$H_3$)$_2$), 1.41 (d, $^3J_{HH}$ = 6.4 Hz, 3 H, CH(C$H_3$)$_2$), 1.22 (d, $^3J_{HH}$ = 6.5 Hz, 3 H, CH(C$H_3$)$_2$), 1.01 (d, $^3J_{HH}$ = 6.6 Hz, 3 H, CH(C$H_3$)$_2$), 0.92 (d, $^3J_{HH}$ = 6.9 Hz, 3 H, CH(C$H_3$)$_2$), 0.90 (d, $^3J_{HH}$ = 7.0 Hz, 3 H, CH(C$H_3$)$_2$), 0.45 (d, $^3J_{HH}$ = 6.7 Hz, 3 H, CH(C$H_3$)$_2$). Due to the low solubility of the sample, no meaningfull 2D NMR spectrum could be recorded. Signals in the aromatic region could not be assigned. $^{13}$C NMR (150.9 MHz, C$_6$D$_6$, 25 °C): δ 168.8, 166.6 (ArNC$C$H$_3$), 137.9, 129.3, 128.6, 125.7, 21,4 (toluene), 150.7, 148.0, 146.6, 146.6, 145.9, 145.6, 143.0, 139.5, 137.0, 134.7, 133.4, 133.1, 131.4, 130.3, 130.2, 129.2, 129.0, 128.9, 127.9, 127.8, 126.9, 126.5, 126.5, 126.3, 126.1, 126.0, 125.4, 124.7, 124.3, 124.1, 124.0, 123.6, 115.1, 110.6 (C$_6$H$_3$-2,6$^i$Pr$_2$ and (CO$C_6$H$_5$)$_2$), 89.0 (γ-$C$H), 29.1, 28.2, 28.1, 28.1 (CH($C$H$_3$)$_2$), 27.4, 26.4, 25.6, 25.5, 24.2, 24.1, 23.4, 23.1, (CH($C$H$_3$)$_2$), 19.1, 17.8 (ArNC$C$H$_3$).

LAl($C_{14}H_{10}O_2$) (**8**): The mother liquor from the synthesis of LAl($C_{14}H_{10}O_2$)$_2$ (**11**) was dried in vacuo and the resulting residue was washed with 20 mL of *n*-hexane, yielding 40 mg of LAl($C_{14}H_{10}O_2$). The washing liquid was concentrated to 10 mL and stored at −30 °C to give a second fraction of LAl($C_{14}H_{10}O_2$). Yield: 55 mg (84 μmol, 37%).

Anal. Calcd. for $C_{43}H_{51}AlN_2O_2$: C, 78.87, H, 7.85; N, 4.28; Found: C, 78.5, H, 7.79; N, 4.25%. ATR-IR: υ 3056, 3018, 2962, 2926, 2868, 1585, 1535, 1462, 1442, 1384, 1316, 1253, 1176, 1105, 1055, 1022, 925, 912, 900, 803, 790, 769, 759, 736, 695, 643, 546, 507, 447, 416 cm$^{-1}$. $^1$H NMR (400 MHz, $C_6D_6$, 25 °C): δ 7.48 (d, $^3J_{HH}$ = 7.8 Hz, 4 H, $C_6H_5$), 7.02 (m, 6 H, $C_6H_3$-2,6$^i$Pr$_2$), 6.99 (m, 4 H, $C_6H_5$), 6.86 (tt, $^3J_{HH}$ = 7.2 Hz, $^4J_{HH}$ = 1.4 Hz, 2 H, $C_6H_5$), 5.03 (s, 1 H, γ-C*H*), 3.35 (sept, $^3J_{HH}$ = 6.7 Hz, 4 H, C*H*(CH$_3$)$_2$), 1.55 (s, 6 H, ArNCC*H$_3$*), 1.44 (d, $^3J_{HH}$ = 6.8 Hz, 12 H, CH(C*H$_3$*)$_2$), 1.08 (d, $^3J_{HH}$ = 6.7 Hz, 12 H, CH(C*H$_3$*)$_2$). $^{13}$C NMR (100.6 MHz, $C_6D_6$, 25 °C): δ 172.5 (ArN*C*CH$_3$), 139.7 (*C*OAl), 138.8, 127.9, 127.6, 125.6 ($C_6H_5$), 144.4, 138.0, 128.2, 124.6 (Ar*C*), 98.2 (γ-CH), 28.9 (*C*H(CH$_3$)$_2$), 25.0, 24.7 (CH(*C*H$_3$)$_2$), 23.2 (ArNCC*H$_3$*).

### 4.9. Synthesis of LGa($C_{14}H_{10}O_2$) (**9**)

A total of 100 mg of LGa (205 μmol) and 43.1 mg of benzil (205 μmol) were combined with 2 mL of benzene and the mixture was stirred overnight, during which the product precipitated as an orange crystalline solid. Yield: 90 mg (129 μmol, 63%).

Anal. Calcd. for $C_{43}H_{51}GaN_2O_2$: C, 74.03, H, 7.37; N, 4.02; Found: C, 73.6, H, 7.37; N, 4.15%. ATR-IR: υ 2962, 2927, 2865, 1585, 1533, 1491, 1464, 1442, 1384, 1318, 1260, 1180, 1105, 1065, 1051, 1022, 929, 912, 803, 765, 724, 699, 683, 627, 535, 481, 445 cm$^{-1}$. $^1$H NMR (400 MHz, $C_6D_6$, 25 °C): δ 7.47 (d, $^3J_{HH}$ = 7.8 Hz, 4 H, $C_6H_5$), 7.01 (m, 10 H, $C_6H_5$, and $C_6H_3$-2,6$^i$Pr$_2$), 6.87 (tt, $^3J_{HH}$ = 7.1 Hz, $^4J_{HH}$ = 1.4 Hz, 2 H, $C_6H_5$), 4.90 (s, 1 H, γ-C*H*), 3.35 (sept, $^3J_{HH}$ = 6.8 Hz, 4 H, C*H*(CH$_3$)$_2$), 1.54 (s, 6 H, ArNCC*H$_3$*), 1.46 (d, $^3J_{HH}$ = 6.7 Hz, 12 H, CH(C*H$_3$*)$_2$), 1.09 (d, $^3J_{HH}$ = 6.9 Hz, 12 H, CH(C*H$_3$*)$_2$). $^{13}$C NMR (100.6 MHz, $C_6D_6$, 25 °C): δ 172.1 (ArN*C*CH$_3$), 139.6 (*C*OGa), 139.3, 128.6, 127.6, 125.4 ($C_6H_5$), 144.1, 138.5, 128.6, 124.5 (Ar*C*), 96.4 (γ-CH), 28.9 (*C*H(CH$_3$)$_2$), 24.8, 24.6 (CH(*C*H$_3$)$_2$), 23.4 (ArNCC*H$_3$*).

### 4.10. Synthesis of LIn($C_{14}H_{10}O_2$)·MeCN (**10**)

A total of 100 mg of LIn (188 μmol) and 39.4 mg of benzil (188 μmol) were balanced in a Schlenk flask and cooled to −80 °C. 5 mL of *n*-hexane was added to the solids, and the mixture was stirred overnight and allowed to warm to room temperature. All volatiles were removed under reduced pressure, the residue was dissolved in a mixture of 1 mL benzene and 10 mL acetonitrile and concentrated to about 1 mL. **10** was obtained as an orange crystalline solid (suitable for sc-XRD) after storage at 6 °C. Yield: 45 mg (61 μmol, 32%).

ATR-IR: υ 2962, 2924, 2868, 2296, 2268, 1592, 1519, 1435, 1384, 1356, 321, 1270, 1178, 1133,1051, 1021, 924, 906, 863, 800, 757, 711, 693, 665, 609, 530, 439 cm$^{-1}$. $^1$H NMR (400 MHz, thf-d$_8$, 25 °C): δ 7.24-7.12 (m, 6 H, $C_6H_3$-2,6$^i$Pr$_2$), 6.90 (d, $^3J_{HH}$ = 7.0 Hz, 4 H, $C_6H_5$), 6.74 (d, $^3J_{HH}$ = 7.4 Hz, 4 H, $C_6H_5$), 6.70 (t, $^3J_{HH}$ = 7.2 Hz, 2 H, $C_6H_5$), 5.19 (s, 1 H, γ-C*H*), 3.32 (sept, $^3J_{HH}$ = 6.9 Hz, 4 H, C*H*(CH$_3$)$_2$), 1.94 (s, 2.5 H, MeCN), 1.84 (s, 6 H, ArNCC*H$_3$*), 1.25 (d, $^3J_{HH}$ = 6.8 Hz, 12 H, CH(C*H$_3$*)$_2$), 1.23 (d, $^3J_{HH}$ = 6.9 Hz, 12 H, CH(C*H$_3$*)$_2$). $^{13}$C NMR (100.6 MHz, thf-d$_8$, 25 °C): δ 172.5 (ArN*C*CH$_3$), 139.7 (*C*OIn), 143.8, 143.3, 127.1, 124.4 (Ar*C*), 142.9, 128.6, 126.9, 123.8 ($C_6H_5$), 96.9 (γ-CH), 28.8 (*C*H(CH$_3$)$_2$), 25.0, 24.5 (CH(*C*H$_3$)$_2$), 24.5 (ArNCC*H$_3$*), 117.3, 0.4 (MeCN). 100 mg LGa

### 4.11. Synthesis of (LAl)$_2$($C_{14}H_{10}O_2$) (**12**)

A total of 23.6 mg of benzil (113 μmol) dissolved in 1 mL of toluene was added dropwise to a solution of 100 mg of LAl (225 μmol) in 1 mL of toluene. The mixture was stirred for 2 days, after which an orange precipitate formed. The suspension was stored overnight at −6 °C. 40 mg of **12** was obtained as an orange powder by filtration. The filtrate consists mainly of LAl and (LAl)($C_{14}H_{10}O_2$) (**8**). Yield: 40 mg (36 μmol, 33%).

Anal. Calcd. for $C_{72}H_{92}Al_2N_4O_2$: C, 78.65, H, 8.43; N, 5.10; Found: C, 78.9, H, 8.51; N, 4.90%. ATR-IR: $\upsilon$ 2964, 2928, 2865, 1529, 1462, 1440, 1384, 1316, 1253, 1179, 1156, 1102, 1061, 1022, 972, 937, 899, 873, 800, 786, 774, 759, 734, 693, 618, 532, 480, 445, 422 cm$^{-1}$. $^1$H NMR (400 MHz, $C_6D_6$, 25 °C): δ 8.53 (d, $^3J_{HH}$ = 7.9 Hz, 2 H, $C_6H_5$), 7.42 (dd, $^3J_{HH}$ = 7.0, 8.4 Hz, 2 H, $C_6H_5$), 7.24–7.11 (m, 8 H, $C_6H_3$-2,6$^i$Pr$_2$), 7.07 (t, $^3J_{HH}$ = 7.3 Hz, 1 H, $C_6H_5$), 7.00 (m, 4 H, $C_6H_3$-2,6$^i$Pr$_2$), 4.96 (s, 1 H, γ-C*H*), 4.88 (dd, $^3J_{HH}$ = 6.0, 8.4 Hz, 2 H, AlC(CHC*H*)$_2$CHAl), 4.84 (s, 1 H, γ-C*H*), 4.02 (d, $^3J_{HH}$ = 8.4 Hz, 2 H, AlC(C*H*CH)$_2$CHAl), 3.45 (sept, $^3J_{HH}$ = 6.7 Hz, 2 H, C*H*(CH$_3$)$_2$), 3.33–3.20 (m, 4 H, C*H*(CH$_3$)$_2$), 3.07 (sept, $^3J_{HH}$ = 6.7 Hz, 2 H, C*H*(CH$_3$)$_2$), 1.92 (t, $^3J_{HH}$ = 6.0, 1 H, AlC(CHCH)$_2$C*H*Al), 1.53 (s, 6 H, ArNCC*H*$_3$), 1.52 (s, 6 H, ArNCC*H*$_3$), 1.33 (d, $^3J_{HH}$ = 6.8 Hz, 6 H, CH(C*H*$_3$)$_2$), 1.30 (d, $^3J_{HH}$ = 6.7 Hz, 6 H, CH(C*H*$_3$)$_2$), 1.25 (d, $^3J_{HH}$ = 7.0 Hz, 6 H, CH(C*H*$_3$)$_2$), 1.16 (d, $^3J_{HH}$ = 6.8 Hz, 6 H, CH(C*H*$_3$)$_2$), 1.12 (d, $^3J_{HH}$ = 6.8 Hz, 6 H, CH(C*H*$_3$)$_2$), 0.96 (d, $^3J_{HH}$ = 6.7 Hz, 6 H, CH(C*H*$_3$)$_2$), 0.92 (d, $^3J_{HH}$ = 6.78 Hz, 6 H, CH(C*H*$_3$)$_2$), 0.82 (d, $^3J_{HH}$ = 6.7 Hz, 6 H, CH(C*H*$_3$)$_2$). $^{13}$C NMR (100.6 MHz, $C_6D_6$, 25 °C): δ 170.8, 170.3 (ArNC*C*H$_3$), 148.5, 132.7 (COAl), 140.5, 126.9, 126.9, 123.0 ($C_6H_5$), 128.6, 123.3 (Al*C*(CHCH)$_2$CHAl), 146.2, 145.3, 143.3, 142.5, 141.4, 141.3, 127.4, 127.1, 125.3, 124.7, 123.8, 123.2 (Ar*C*), 98.1, 97.4 (γ-CH), *39.6, 32.7* (Al*C*(CHCH)$_2$*C*HAl), 30.0, 28.9, 28.5, 28.2 (*C*H(CH$_3$)$_2$), 26.6, 26.5, 25.0, 24.9, 24.8, 24.6, 23.8, 23.6 (CH(*C*H$_3$)$_2$), 23.8 (ArNC*C*H$_3$). Peaks in italic were only observed in HSQC or HMBC respectively.

### 4.12. Synthesis of L'B(C$_{14}$H$_{10}$O$_2$) (**13**)

A total of 56.9 mg of KC$_8$ (421 μmol) and 41.1 mg of benzil (198 μmol) were cooled to −80 °C, 5 mL of thf was added and the resulting mixture was warmed to room temperature upon stirring. The liquid phase was transferred through a filter cannula into a cooled (−80 °C) Schlenk flask containing 100 mg of L'BCl$_2$ (198 μmol). All volatiles were removed at room temperature in a vacuo, and the resulting residue was extracted with 5 mL of toluene. The extract was concentrated and stored at −30 °C, yielding 20 mg of an orange crystalline solid. The mother liquor was further concentrated to give a second fraction. Yield: 30 mg (46 μmol, 24%).

Anal. Calcd. for $C_{31}H_{17}BF_{10}N_2O_2$: C, 57.26, H, 2.64; N, 4.31; Found: C, 57.4, H, 2.63; N, 4.31%. ATR-IR**:** $\upsilon$ 1564, 1508, 1469, 1450, 1392, 1290, 1264, 1130, 1084, 1065, 1042, 1022, 987, 925, 871, 761, 724, 693, 663, 619, 597, 523, 457 cm$^{-1}$. $^1$H NMR (400 MHz, $C_6D_6$, 25 °C): δ 7.62–7.57 (m, 4 H, $C_6H_5$), 6.98 (t, $^3J_{HH}$ = 7.8 Hz, 4 H, $C_6H_5$), 6.86 (t, $^3J_{HH}$ = 7.41 Hz, 2 H, $C_6H_5$), 4.69 (s, 1 H, γ-C*H*), 1.26 (s, 6 H, ArNCC*H*$_3$). $^{11}$B NMR (128.4 MHz, $C_6D_6$, 25 °C): δ 7.0. $^{13}$C NMR (100.6 MHz, $C_6D_6$, 25 °C): δ 166.2 (ArNC*C*H$_3$), 136.7 (COB), 133.4, 128.4, 127.1, 126.1 ($C_6H_5$), 97.6 (γ-CH), 20.6 (ArNC*C*H$_3$). $^{13}$C {$^{19}$F} NMR (150.9 MHz, $C_6D_6$, 25 °C): δ 144.4, 141.0, 137.9, 116.4 ($C_6F_5$). $^{19}$F NMR (376.5 MHz, $C_6D_6$, 25 °C): δ −143.5 (d, $^3J_{HH}$ = 17.6 Hz, 4 F, $C_6F_5$), −153.8 (t, $^3J_{HH}$ = 22.5 Hz, 2 F, $C_6F_5$), −162.3 (m, 4 F, $C_6H_5$).

### 4.13. Synthesis of L'BH$_2$ (**14**)

At −80 °C, 783 μL (783 μmol) of a 1 M (*n*-hexane) L-selectride solution was added to 200 mg of L'BCl$_2$ (391 μmol) dissolved in 10 mL of toluene. The reaction mixture was allowed to reach ambient temperature overnight. All volatiles were then removed in vacuo, and the resulting residue was extracted with 10 mL of *n*-pentane. The extract was concentrated and stored at −30 °C to give 70 mg of **14** as a yellow crystalline solid. Yield: 70 mg (158 μmol, 41%).

Anal. Calcd. for $C_{17}H_9BF_{10}N_2$: C, 46.19, H, 2.05; N, 6.34; Found: C, 46.6, H, 2.32; N, 6.16%. ATR-IR: $\upsilon$ 2964, 2428, 2370, 2291, 2192, 1582, 1549, 1504, 1467, 1438, 1392, 1344, 1260, 1206, 1069, 1051, 1024, 1009, 989, 939, 817, 788, 782, 726, 647, 561, 472 cm$^{-1}$. $^1$H NMR (400 MHz, $C_6D_6$, 25 °C): δ 4.77 (s, 1 H, γ-C*H*), 3.77 (s (br), 1 H, B*H*$_2$), 3.46 (s (br), 1 H, B*H*$_2$), 1.24 (s, 6 H, ArNCC*H*$_3$). $^{11}$B NMR (192.5 MHz, $C_6D_6$, 25 °C): δ 7.0 (*B*). $^{13}$C NMR (100.6 MHz, $C_6D_6$, 25 °C): δ 167.6 (ArNC*C*H$_3$), 143.7 (m, $^2J_{HH}$ = 250 Hz, $C_6F_5$), 140.3 (m, $^2J_{HH}$ = 252 Hz, $C_6F_5$), 138.2 (m, $^2J_{HH}$ = 252 Hz, $C_6F_5$), 120.1 (td, $J_{HH}$ = 17.8, 5.2 Hz, $C_6F_5$), 99.6 (γ-CH), 19.8 (ArNC*C*H$_3$). $^{19}$F NMR (376.5 MHz, $C_6D_6$, 25 °C): δ −143.7 (d, $^3J_{HH}$ = 20.8 Hz, 4 F, $C_6F_5$), −156.2 (t, $^3J_{HH}$ = 22.3 Hz, 2 F, $C_6F_5$), −161.9 (td, $J_{HH}$ = 23.3, 6.1 Hz, 4 F, $C_6H_5$).

*4.14. Crystallographic Details*

Crystals were mounted on nylon loops in inert oil. Data of were collected on a Bruker AXS D8 Kappa diffractometer (**2, 5, 7, 9, 10, 11, 12b**) with APEX2 detector (monochromated Mo$_{Ka}$ radiation, $\lambda$ = 0.71073 Å) and on a Bruker AXS D8 Venture diffractometer (**1, 4, 6, 8, 12a, 13**) with Photon II detector (monochromated Cu$_{Ka}$ radiation, $\lambda$ = 1.54178 Å, microfocus source) at 100(2) K. Structures were solved by Direct Methods (SHELXS-2013) [40] and refined anisotropically by full-matrix least-squares on $F^2$ (SHELXL-2014) [41–43]. Absorption corrections were performed semi-empirically from equivalent reflections based on multi-scans (Bruker AXS APEX2). Hydrogen atoms were refined using a riding model or rigid methyl groups.

**1**: An isopropyl group is disordered over two positions. C24 and C24′ were refined with common positions and displacement parameters (EXYZ, EADP). RIGU restraints were applied to the displacement parameters of the disordered atoms.

**2**: Two isopropyl groups are disordered over two positions. Their bond lengths and angles were restrained to be equal (SADI) and RIGU restraints were applied to their displacement parameters. Additional SIMU restraints were applied to the group in residue 2. In residue 1, the diol ligand is disordered over two positions. All corresponding bond lengths were restrained to be equal (SADI) and RIGU restraints were used to refine the displacement parameters. Atoms in close proximity (O4, O4′ and C33, C33′) were refined with common displacement parameters (EADP). One solvent molecule is disordered over two positions. All bond lengths and angles of the solvent molecules were restrained to be equal (SADI) and the molecule was restrained to be planar (FLAT). RIGU restraints were applied to the displacement parameters of the solvent atoms. The quantitative results of the disordered moieties should be scrutinized, and especially the data for the diol ligand may be unreliable. In addition, ice formed during the measurement and combined with a scattering of the mount/oil used as glue, clearly visible background scattering was found in the frames. This leads to some distorted intensities. The resolution of (090) could be related to the (103) reflection of water and was therefore omitted. Three reflections shaded by the beamstop were also ignored in the refinement. The remaining most disagreeable reflections have a resolution of >2 Å with $F_{obs}$ higher than $F_{calc}$. **2** was refined as a 2-component inversion twin.

**5**: The crystal was a non-merohedral twin and the model was refined against de-twinned HKLF4 data. At low angles, some disagreeable reflections are found. These are either due to poor separation of overlaps in the integration or, more likely, to background scattering caused by the icing of the crystal. Three reflections shaded by the beam-stop were ignored in the refinement (OMIT).

**6**: The structure contains a 2-methyltetrahydrofuran molecule highly disordered over an inversion center. The final refinement was conducted with a solvent-free data set from a PLATON/SQUEEZE run [44]. The molecule was included in the sum formula for completeness.

**7**: An isopropyl group is disordered over two positions. Its bond lengths and angles were restrained to be equal (SADI) and RIGU restraints were applied to its displacement parameters. The structure contains two highly disordered *n*-hexane molecules. The final refinement was conducted with a solvent-free data set from a PLATON/SQUEEZE run [44]. The molecules were included in the sum formula for completeness.

**9**: A diisopropylphenyl group is disordered over two positions. RIGU restraints were applied to the displacement parameters of the corresponding atoms.

**12a**: A benzene molecule is disordered over two positions. RIGU restraints were applied to the displacement parameters of the atoms of the solvent molecules.

**12b**: An *n*-hexane molecule is disordered over a center of inversion. The bond lengths and bond angles of all solvent molecules were restrained to be equal and RIGU restraints were applied to the displacement parameters of their atoms. An additional SIMU restraint was used for the disordered molecule on the center of inversion. Its displacement parameters suggested further disorder that could not be resolved any further.

**Supplementary Materials:** The following supporting information can be downloaded at: https://www.mdpi.com/article/10.3390/chemistry5020064/s1, Heteronuclear NMR ($^1$H, $^{11}$B, $^{13}$C, 19F) and IR data of all compounds (Figures S1–S48) as well as crystallographic details (Table S1a–d and Figures S53–S57).

**Author Contributions:** Conceptualization, H.M.W. and S.S.; investigation, H.M.W.; sc-XRD acquisition and refinement, C.W.; writing—original draft preparation, H.M.W.; writing—review and editing, S.S.; supervision, S.S. All authors have read and agreed to the published version of the manuscript.

**Funding:** This work was supported by the Deutsche Forschungsgemeinschaft DFG (INST 20876/282-1 FUGG) and the University of Duisburg-Essen.

**Data Availability Statement:** Spectroscopic data and crystallographic details are given in the electronic supplement. The structures of compounds **1**, **2,** and **4–13** in the solid state were determined by single-crystal X-ray diffraction and the crystallographic data have been deposited with the Cambridge Crystallographic Data Centre as supplementary publication nos. CCDC-2251780 (**1**), -2251781 (**2**), -2251782 (**4**), -2251783 (**5**), -2251784 (**6**), -2251785 (**7**), -2251786 (**8**), -2251787 (**9**), -2251788 (**10**), -2251789 (**11**), -2251790 (**12a**), -2251791 (**12b**), and -2251909 (**13**). Copies of the data can be obtained free of charge on application to CCDC, 12 Union Road, Cambridge, CB21EZ (fax: (+44) 1223/336033; e-mail: deposit@ccdc.cam-ak.uk).

**Acknowledgments:** We are thankful to A. Gehlhaar and Y. Schulte (University of Duisburg-Essen) for assistance with the sc-XRD data acquisition, to T. Schaller (University of Duisburg-Essen) for performing non-routine NMR experiments, and to J. Tewes (University of Duisburg-Essen) for providing L'BCl$_2$.

**Conflicts of Interest:** The authors declare no conflict of interest.

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
