# Peer review of "Synthesis of 5-Metalla-Spiro[4.5]Heterodecenes by [1,4]-Cycloaddition Reaction of Group 13 Diyls with 1,2-Diketones"

_chemistry, doi:10.3390/chemistry5020064_

Round 1

Reviewer 1 Report

This study reveals describes a useful and informative exploration of the [4+1] cycloaddition behavior of group 13 diyls with 1,2-dicarbonyl compounds. The description of the chemistry is satisfactory and the characterization of the  reaction products is performed through spectral analysis and X-ray crystallography. The provided spectra suggest the products have good purity. The work clearly contributes to the field by demonstrating new chemical behavior and some of the limitations of it.

This work is worthy of publication, but I think some corrections and presentation improvement would create a stronger paper.

 1.     The introduction would benefit from a graphic showing some of the general structures  that are discussed. This would include some compounds from other researchers.  

2.     2. The manuscript would benefit from the listing of chemical yields as part of the reactions schemes in the R and D section.

3.     3. Is there an explanation for the asymmetry of 6 in the 1H NMR? Compound 6 is found to be dimeric in the solid state. However, the 1H NMR of 6 does not hold the clean symmetry of compounds 4 and 5. I think this merits comment about the possible solution structure of 6.

4.     4. In the experimental, this formula is presented for compound 1: 

C122 H106Sb2 F48Ga2N4

          This needs to be corrected.

5. The manuscripts needs a proof-read for typos. Here are two to repair:

-On page 4 of the manuscript, what is an    α,β-ketone

-Bicyclus  should probably be bicycle

6. At one point the following is written: 13C NMR (100.6 MHz, C6 D6, 25 °C): many resonances were not observed due to line broadening (cp. 1H NMR spectrum).

Part of the problem is the lack of adequate compound in the NMR tube. Maybe the weak spectrum is also due to solubility problems as the authors identify elsewhere in the manuscript.

Reviewer 2 Report

In their manuscript Schulz et al. provided interesting results on the reaction of monovalent group 13 (Al, Ga, In, Tl) compounds with 1,2-diketones. The latter were exemplified by 2,3-butanedione, acenaphthoquinone, and benzil (1,2-diphenyl-ethane-1,2-dione). Structure of nearly all products was confirmed by X-ray analysis. The manuscript can be published in Chemistry after minor (though extensive revision). Particularly, the Authors are encouraged to check the manuscript for misprints. I.e., sometimes Ga is confused with Al and so on. Some of such misprints are mentioned below.

 Main remarks:

-        The Reviewer believes that naming of compounds (in the Title and so on) as “5-Metalla-spiro[4.5]decenes” is misleading, since the “decene” backbone possesses not only metal, but also oxygen and nitrogen atoms. Please, revise.

-        Formatting of the Schemes should be unified. E.g., now the size of structures in Schemes 3 and 5 is different. And so on.

-        Please, show yields and reaction conditions in the Schemes.

-        Full structure of dipp ligand L is better to be clearly shown in the beginning of the manuscript (e.g. in Scheme 1).

-        It’s advisable to show structure of compounds at the copies of NMR.

-        Integrals should be shown at the copies of 1H NMR.

-        The Authors reported reactions with 1,2-diketones. Could the Authors comment on the possible reaction (e.g., literature references, if it has been already done) between compounds 1 and mono-carbonyl compounds? Is pinacol-type coupling possible?

-        Similar to the previous question. Could the authors provide data (literature references, if it has been already done) on the reaction with such 1,2-dicarbonyl compounds as o-benzoquinones?

-        Could the Authors provide more data on the transformation LGa->1->2? Particularly, why the preparation of 1 is possible? How selective is it? Is it because of difference in reaction conditions necessary for LGa->1 and 1->2?

-        Scheme 4. Please, provide possible reaction mechanism for the formation of product 12.

-        Line 233. Where did Sb and F come to the brutto-formula of 1?

-        Line 258. Should it be C11H12O3?

-        Line 280. Al in the title but LGa in the procedure as well as Ga in the brutto-formula. In Scheme 2 compound 4 is drawn to be with Al. Please, check.

-        Line 317. Compound is claimed to be with In, but Ga is shown in brutto-formula. Please, check.

-        Line 330. Benzil is mentioned, while acenaphthoquinone in Scheme 2. Please, check.

 Other remarks:

-        Line 10. Use of “M = Al, Ga, In, Tl” is better than “M = Al - Tl”.

-        Keyword “low-valent” looks unclear. Is “low-valent metal” better?

-        Check Figure captions. E.g., is Ga missed in Fig. S3? ddp is mentioned in Fig. S19, while this abbreviation is not decoded in the manuscript.

-        Line 62, “previously reported higher selectivity of LGa compared to LAl”. Please, provide appropriate reference.

-        Scheme 1. benzaldehyde, not benzaldehyd.

-        Line 73, “acenaphthenequinone”, while “acenaphthoquinone” in Scheme 2. Please, unify the name of the compound.

-        Line 71, “[1,4]-cycloaddition”, while line 82 “(1,4) cycloaddition”. And so on. Please, unify.

-        Scheme 5. Selectride, not Selektride.

-        Line 221. Was 13C really referenced to C6D5H and not C6D6? The same for THF-d7/d8.

-        Line 336. formation, not formatiopn.

-        Line 345. What is ddp?
